# Population-Based Real-World Outcomes of Post-Operative Adjuvant Brain Cavity Radiotherapy Versus Observation

**DOI:** 10.3390/curroncol32060345

**Published:** 2025-06-11

**Authors:** Zhang Hao (Jim) Li, Linden Lechner, Jennifer Wang, Nan Hui (Susan) Yao, Andrew Lee, Serge Makarenko, Mostafa Fatehi, Herve H. F. Choi, Ermias Gete, Fred Hsu, Waseem Sharieff, Shrinivas Rathod, Hannah Carolan, Jessica Chan, Roy Ma, Alan Nichol, Thi Nghiem, Justin Oh

**Affiliations:** 1Division of Radiation Oncology, Department of Surgery, University of British Columbia, Vancouver, BC V5Z 4E6, Canadawaseem.sharieff@bccancer.bc.ca (W.S.); anichol@bccancer.bc.ca (A.N.); 2Faculty of Medicine, University of British Columbia, Vancouver, BC V6T 1Z3, Canada; 3Division of Neurosurgery, Department of Surgery, University of British Columbia, Vancouver, BC V5Z 1M9, Canada; 4Department of Medical Physics, BC Cancer Agency, Vancouver, BC V5Z 4E6, Canada; 5Data and Analytics, BC Cancer Agency, Vancouver, BC V5Z 4E6, Canada

**Keywords:** stereotactic radiotherapy, stereotactic radiosurgery, palliative radiotherapy, CNS metastasis, radionecrosis, leptomeningeal disease

## Abstract

This study looked at all patients in British Columbia who had surgery to remove brain metastases between 2018 and 2020. Researchers compared those who received subsequent targeted radiation (stereotactic radiotherapy) with those who were only monitored afterward. The results indicate that those who received stereotactic radiotherapy were at significantly reduced risk of the cancer returning at the original surgical site after one year (69% control with stereotactic radiotherapy versus 31% with observation). However, it did not affect patients’ overall survival or the risk of the cancer spreading elsewhere in the brain. Patients who had complete tumor removal and received post-operative stereotactic radiotherapy had the best outcomes. Side effects of radiation treatment were generally tolerable. These population-based real-world results support current clinical guidelines recommending stereotactic radiotherapy to improve local control after brain metastasis surgery.

## 1. Introduction

Brain metastases are the most common type of malignant central nervous system tumor [1]. They can cause debilitating neurological symptoms impairing patients’ quality of life. Older studies show that 8–10% of all cancer patients will develop brain metastases during their lifetime [2,3]. This number is thought to be increasing, in part due to greater availability of improved imaging techniques and more effective systemic treatments [4]. For certain cancer histologies, such as lung adenocarcinoma, small cell lung cancer, and melanoma, about a quarter of patients with metastatic disease present with de novo brain metastases at the time of diagnosis, further highlighting the importance of optimal treatment strategies for this cohort of patients [5].

For patients with symptomatic brain metastases, the traditional approach was metastasis-direct therapy with surgery and adjuvant whole-brain radiotherapy (WBRT) with or without a subsequent radiosurgery boost [6,7]. A randomized controlled trial (RCT) has shown that adjuvant stereotactic radiosurgery (SRS) and stereotactic fractionated radiotherapy (SFRT) offer greater local control compared to observation [8]. When compared to WBRT, SRS/SFRT has similar local control, but with less neurocognitive deterioration [9,10,11]. While certain drawbacks of SRS/SFRT exist, such as the risk of radionecrosis and nodular leptomeningeal disease [8,9,10,12], the standard of care has shifted to SRS/SFRT in the post-operative setting since 2021 [13,14,15,16].

Population-based studies examining how these updated guidelines translate into real-world outcomes are lacking in the literature. Because patients are frequently excluded from RCTs due to factors such as age, comorbidities, and polypharmacy [17], RCT patients may not necessarily reflect the heterogeneity of the general population. Therefore, integrating population-based studies with RCTs can enhance the inclusivity of the existing evidence. Notably, many patients enrolled in landmark RCTs demonstrating the local control benefit of post-operative SRS/SFRT were treated prior to the widespread adoption of immunotherapies and targeted agents with proven intracranial efficacy—such as osimertinib, alectinib, ipilimumab, and nivolumab [18,19,20]. Therefore, it is essential to evaluate whether the local control advantages of SRS/SFRT persist in the contemporary treatment landscape shaped by these novel systemic therapies. Although various large-scale retrospective studies have been conducted on adjuvant SRS/SFRT, their patient cohorts tend to be treated at a time before randomized trials were published, comparing SRS/SFRT to WBRT and observation, and often there was no comparison group of patients receiving WBRT or observation [21,22,23,24]. Without updated real-world studies, it is difficult to ascertain how the new guidelines translate into practical benefits for the general population. Therefore, this study endeavors to examine the clinical outcomes of patients within British Columbia—a province of 5 million people—who underwent brain metastasectomy followed by either SRS, SFRT, or observation.

## 2. Materials and Methods

A multi-center retrospective review was performed on all patients within British Columbia who were diagnosed with secondary malignant neoplasm of the brain and cerebral meninges (ICD-10 code: C79.3) and received brain metastasectomy between January 2018 and December 2020. Patients were identified through the provincial Ministry of Health Discharge Abstract Database by selecting for those with an ICD-10 diagnosis code of C79.3 and a documented history of neurosurgical intervention. The year 2018 was chosen as the initial cut-off point, as it follows the publication of RCTs demonstrating a local control benefit of post-operative SRS/SFRT compared to observation [8]. Patients treated after 2018 may also have been eligible for newer systemic therapies with enhanced intracranial efficacy [18,19,20]. The year 2020 was selected as the end point, as it precedes the release of consensus guidelines formally recommending the use of post-operative SRS/SFRT [13,14]. This interval represents a transitional period during which we would expect high variability in physician adoption of post-operative SRS/SFRT, which provides a unique opportunity to examine the real-world outcomes of post-operative SRS/SFRT compared to observation.

Referrals to radiation oncology, decisions to offer radiotherapy, radiotherapy planning, and follow up schedules were arranged according to the clinical discretion of the treating physicians. Patients were included in our study if they received surgical resection of brain metastases. Patients were excluded if they received WBRT, a total radiotherapy dose of less than 15 Gy, or if radiotherapy was delivered in more than 5 fractions. Using the time of surgery as a baseline, patient characteristics were collected. The primary study endpoint was local control (LC), which was defined as time from surgery to radiographic evidence of any new or progressive contrast-enhancing nodularity contiguous to or within the tumor cavity [8], as noted by the reporting radiologist. In the case of any ambiguity, the date of the first ambiguous imaging was used to calculate local recurrence if it was later proven radiographically or pathologically. Cases of radiographic recurrence, which were later biopsy-proven negative, did not count as a local recurrence. Patients were censored at the date of last brain imaging if they did not experience local recurrence at the date of death or last follow up. Secondary endpoints included distant intracranial control (DICC), leptomeningeal disease (LMD), radionecrosis, overall survival (OS), and time to change in systemic therapy. DICC was defined as the time from surgery to the appearance of a new or progressive lesion on imaging that was distinct from the tumor cavity. LMD was assessed based on radiology reports, and when available, cerebrospinal fluid analysis [9]. Radionecrosis was evaluated using radiology reports and supplemented by pathology findings, if applicable [23]. In situations where distinction between local recurrence and radionecrosis was unclear, the cases were reviewed at multidisciplinary tumor board meetings, and the final diagnosis would be determined by group consensus. Cause of death was categorized as neurological versus non-neurological. Patients who died with progressive or severe neurologic dysfunction were considered to have died of neurologic causes, regardless of whether their systemic cancer or any intercurrent disease(s) were also contributory; deaths were non-neurologic if the patient was stable or improving from a neurologic perspective at the time of death [25]. Time to change in systemic therapy was calculated as the time from surgery to initiation or switch in the systemic agent. Toxicity from radiotherapy was retrospectively evaluated from the treating physician’s documentation in accordance with the Common Terminology Criteria for Adverse Events (CTCAE version 5.0). All cases of grade 3 or higher toxicity were independently reviewed by at least two researchers.

As per institutional standards, patients undergoing SRS were prescribed 18–24 Gy in a single fraction to the planning target volume (PTV); patients being treated with SFRT were prescribed 30–35 Gy in 5 daily fractions to the PTV. The PTV was created from a 1 mm isotropic expansion of the gross tumor volume (GTV), as visualized on post-operative CT/MRI. Treatment plans were optimized to aim for Dmax < 150%, with Dmax situated within the GTV, and PTV_100%_ > 99%. Whenever possible, pre-radiotherapy MRI with ≤1 mm slice thickness was obtained. Radiotherapy was delivered using 6 MV photons with the VMAT technique by linear accelerators. Patients were immobilized with a thermoplastic shell, and image verification with cone beam CT was performed. This is consistent with published guidelines on brain SRS/SFRT [15,26]. Patients were followed with brain MRI every 3–4 months for the first two years, and thereafter every 6 months.

The statistical analysis was performed using SPSS (Version 29.0.1.0). The Kaplan–Meier method was used to calculate LC, DICC, and OS, and curves were compared using the log-rank test. Univariable analysis using a Cox proportional hazards model was performed to evaluate the significance and magnitude of potential predictive factors that may affect the primary and secondary outcomes of this study. Statistical significance was accepted for *p*-values < 0.05. Factors assessed include age at the time of brain metastasis, biological sex, Eastern Cooperative Oncology Group (ECOG) performance status, Karnofsky Performance Scale (KPS), Charlson Comorbidity Index, residual disease after surgery, number of brain metastases, maximum tumor dimension, presence of extracranial disease (ECD), progression of ECD, en bloc tumor resection, receipt of adjuvant SRS or SFRT, and receipt of systemic therapy prior to surgery. En bloc resection was operationalized as the circumferential dissection of the tumor without violating the tumor capsule, as per the operative report [27]. For the multivariable analysis, adjuvant SRS/SFRT and additional prognostic factors with *p*-value < 0.1 from the univariable analysis were placed into a backward stepwise procedure to identify independent predictive factors. Exploratory ad hoc analyses would then be performed to determine if any of these predictive factors could allow for treatment escalation or de-escalation. All procedures were performed in compliance with the Helsinki Declaration, and institutional research ethics board approval was obtained prior to initiation of the study.

## 3. Results

### 3.1. Cohorts and Demographics

From January 2018 to December 2020, 193 patients underwent brain metastasectomy. Of which, 80 patients received WBRT and were excluded. Among the 113 remaining patients, 31 received post-operative SRS or SFRT while 82 did not. Their baseline characteristics are summarized in Table 1. The median time from brain metastasis diagnosis to surgery was 7 days (IQR: 4–26 days) in the SRS/SFRT cohort and 14 days (IQR: 5–41 days) in the observation cohort (*p* = 0.15). The proportion of patients receiving SRS/SFRT versus observation is not statistically correlated to the year in which they were treated (*p* = 0.5; Figure A1).

As shown in Table 1, the only significant baseline difference between the two cohorts was that the observation cohort had smaller tumors (*p* = 0.05). For the post-operative SRS/SFRT cohort, the median time from surgery to radiotherapy start was 46 days (range: 22–95 days). The median BED delivered to the SRS/SFRT cohort was 47.25 Gy_20_ (IQR: 39.0–47.25 Gy_20_). Detailed dosimetric data can be found in Table A1. The reasons why patients in the observation cohort did not receive post-operative SRS/SFRT are listed in Table 2.

### 3.2. Local Control

At a median follow up of 6.6 months (range: 0.2–69.6 months), 10 cases of local recurrence occurred in the SRS/SFRT cohort (32%), compared to 38 cases in the observation cohort (46%). The 12-month local control rates were 69% (95% CI: 50–88%) and 31% (95% CI: 18–45%), respectively (*p* < 0.001), as shown in Figure 1.

Specifically, for patients with non-small cell lung cancer (NSCLC), their 12-month LC was 65% (95% CI: 37–93%) if they received SRS/SFRT, compared to 24% (95% CI: 7–41%) if they went on observation (*p* = 0.001).

### 3.3. Distant Intracranial Control

There were no statistically significant differences in the DICC rates between the two cohorts [Figure 2]. The 12-month DICC was 44% (95% CI: 26–63%) in the SRS/SFRT cohort and 46% (95% CI: 30–62%) in the observation cohort (*p* = 0.9).

### 3.4. Overall and Neurologic Survival

A total of 25 deaths occurred in the SRS/SFRT cohort (81%) and 69 in the observation cohort (84%). The 12-month OS was higher in the SRS/SFRT cohort (61%; 95% CI: 44–78%) than the observation cohort (32%; 95% CI: 22–43%; *p* = 0.03). The median OS was 14 months (95% CI: 10–18 months) and 3.5 months (95% CI: 1.4–5.7 months) in the SRS/SFRT and observation cohorts, respectively. When patients with an ECOG of 2 or higher were excluded from the OS analysis, the 12-month OS was 60% (95% CI: 41–79%) in the SRS/SFRT cohort and 40% (95% CI: 27–52%) in the observation cohort (*p* = 0.29). A total of 15 of 25 deaths in the SRS/SFRT cohort were neurologic (60%), whereas 37 of 69 deaths in the observation cohort were neurologic (54%). Differences in 12-month neurologic survival were not statistically significant: 73% (95% CI: 56–89%) in the SRS/SFRT cohort, compared to 56% (95% CI: 43–68%) in the observation cohort (*p* = 0.2).

### 3.5. Systemic Therapy

At the time of surgery, 4 patients (13%) in the SRS/SFRT cohort and 23 patients (28%) in the observation cohort were on systemic therapy [Table 1]. Specifically, 3 (10%) in the SRS/SFRT cohort and 18 (22%) in the observation cohort were on targeted agents or immunotherapies, such as pembrolizumab, afatinib, and nivolumab. After surgery, 4 more patients (13%) in the SRS/SFRT cohort received targeted therapy or immunotherapy, compared to 16 (20%) in the observation cohort. The proportion of patients who received targeted therapy or immunotherapy after surgery was not statistically different between the two cohorts (*p* = 0.24). The mean time to change in systemic therapy was 41.0 months (95% CI: 26.3–55.7 months) in the SRS/SFRT cohort. This was not significantly different from the observation cohort (31.1 months; 95% CI: 20.3–41.9 months, *p* = 0.18).

### 3.6. Additional Analyses

On univariate analysis of local control, besides post-operative SRS/SFRT, residual disease after surgery and en bloc resection had a *p*-value < 0.1 and were selected for multivariate analysis. Residual disease after surgery, en bloc resection, and post-operative SRS/SFRT were statistically significant on multivariable analysis (<0.05) [Table 3]. Since the observation cohort had smaller tumor dimensions, an additional MVA of the three above variables and tumor dimension was performed; the latter was not significantly predictive of LC (*p* = 0.1; Table A2). Post hoc analyses were performed to explore if there were any differences in LC between the two cohorts when the analysis was restricted to patients who had en bloc resection or no residual disease. For patients who underwent en bloc resection (*n* = 32), the 12-month LC was 53% (95% CI: 13–94%) in the SRS/SFRT cohort and 55% (95% CI: 27–84%) in the observation cohort (*p* = 0.6). Among patients who had no residual disease after resection (*n* = 86), post-operative SRS/SFRT was still predictive of better LC, the 12-month LC being 81% (95% CI: 62–100%) and 34% (19–50%) for SRS/SFRT versus observation, respectively (*p* = 0.002).

Leptomeningeal disease after surgery was observed in 3 patients (10%) in the SRS/SFRT cohort and 2 patients (2%) in the observation cohort (*p* = 0.1). From a toxicity standpoint, 4 patients in the SRS/SFRT cohort developed brain radionecrosis after radiotherapy (13%), of which 2 had asymptomatic grade 1 radionecrosis, and another 2 experienced symptomatic grade 2 radionecrosis. No grade 3 or higher toxicities related to radiotherapy were documented.

The authors acknowledge a risk of bias by categorizing patients who died before their planned SRS/SFRT (*n* = 3) and those who had treatment delays resulting in local recurrence (*n* = 2) in the observation cohort. Therefore, a post hoc sensitivity analysis was performed by analyzing patients based on their intended treatment (pseudo-ITT), in which these 5 patients were moved from the observation cohort into the SRS/SFRT cohort, as the original intent was to treat them with SRS/SFRT prior to death or local recurrence. Additionally, the 17 patients who were unfit for SRS/SFRT were excluded from the pseudo-ITT analysis altogether, due to risk of bias. This results in 36 patients in the pseudo-ITT SRS/SFRT cohort and 60 in the pseudo-ITT observation cohort. The only significant baseline difference between the two pseudo-ITT cohorts is that the SRS/SFRT cohort had larger pre-operative tumor dimensions (*p* = 0.011; Table A3). The 12-month rates for LC, DICC, OS, and neurologic survival (NS) in the pseudo-ITT analysis are shown in Table 4, along with the values from the original per protocol analysis for comparison.

## 4. Discussion

This population-based retrospective study reviewed the clinical outcomes of all patients within British Columbia who underwent brain metastasectomy followed by either post-operative SRS/SFRT or observation. As demonstrated in Table 1, baseline patient characteristics were similar between the two cohorts at the time of surgery. The primary endpoint, local control (LC), was significantly higher in the post-operative SRS/SFRT cohort than the observation cohort, which is in accordance with previously published RCT data [8]. The replication of these findings in a real-world population-based setting lends further support that post-operative radiotherapy is effective at improving LC outcomes outside the context of a randomized trial or select institutional studies. On multivariable analysis, en bloc resection and absence of residual disease after surgery were also predictive of better local control. Post hoc analysis of these factors revealed that among patients with no residual disease, post-operative SRS/SFRT were still associated with greater local control. In addition, for patients who underwent en bloc resection, the sample size (*n* = 32) was too small for any definitive conclusions to be drawn. Notably, the local control benefit of post-operative SRS/SFRT remains consistent even though approximately one-third of patients received targeted or immunotherapy. This supports the continued relevance of current practice guidelines in the context of modern CNS-penetrating systemic treatments.

Interestingly, in our original analysis there was an overall survival (OS) benefit with post-operative SRS/SFRT. However, this is most likely due to selection bias, as the sensitivity analysis revealed no OS benefit after removing the patients who were unfit for SRS/SFRT and recategorizing the patients who died before SRS/SFRT could be delivered. This is supported by the findings that neither the original nor the sensitivity analyses demonstrated a difference in neurological survival (NS) [Table 4], and no OS difference was found after excluding patients with an ECOG of 2 or higher (*p* = 0.29).

The rates of leptomeningeal disease in our study were 10% in the SRS/SFRT cohort and 2% in the observation cohort. 13% of patients in the SRS/SFRT cohort developed grade 1–2 radionecrosis. These are consistent with established values in the literature [21,28]. There were insufficient cases of leptomeningeal disease and radionecrosis for further analysis on predictive factors for these outcomes. Given the retrospective design of our study, prospective toxicity data were not available. However, retrospective review revealed no documented grade 3 or higher toxicities.

Study inclusion years between 2018 and 2020 were specifically chosen to compare the outcome between post-operative SRS/SRT and observation while minimizing baseline characteristic discrepancies between the two subsets and to allow for sufficiently long follow up. Between 2018 and 2020, uptake of SRS/SFRT among treating physicians was variable as the guidelines recommending routine post-op SRS was published after 2020 [13,14,15]. The most common reason for omission of post-operative SRS/SFRT was physician preference (44%). Another potential reason for physician hesitancy may be due to the lack of OS benefit and reported quality-of-life outcomes in randomized evidence [8]. These considerations may be especially pertinent in real-world settings, where resource limitations pose an issue. From an institutional standpoint, radiation oncology was not consulted in 16% of cases in the observation cohort. This indicates an area of potential improvement, where institutional protocols could be optimized to ensure that patients could receive the necessary referrals.

Compared to other retrospective studies published on post-operative SRS/SFRT, strengths of our study include having population-based data and a large observation cohort for comparison. These offer valuable insight into the true benefits of SRS/SFRT on a population level and generalizability of the guideline recommendation. One limitation to our study is the small sample of SRS/SFRT cases which limits analyses stratified for performance status and tumor histology, and detailed dosimetric analysis to evaluate additional predictive factors for local control. Similarly, the numbers of cases with leptomeningeal recurrence and radionecrosis were low and not sufficient for any exploratory analyses. In addition, as this was a retrospective study, another limitation is the absence of quality-of-life data. This precludes assessment of patient-reported effects of SRS/SFRT. Understandably, without data demonstrating improved quality-of-life or a significant neurologic survival benefit, some patients may not wish to receive SRS/SFRT, and some physicians may not feel inclined to offer it. We acknowledge that a sizeable number of patients (*n* = 80) who received WBRT was excluded. Because randomized evidence has shown worse cognition and no OS benefit with WBRT, practice guidelines have moved away from it in this context [13,14,15]. Hence a comparison between WBRT and SRS/SFRT was not within the scope of our current study.

Directions for future research include performing prospective studies examining quality-of-life data for patients, collaborating with other centers to gather a more statistically robust sample size for dosimetric analyses, and to identify potential subgroups for which post-operative SRS/SFRT could be safely omitted. Finally, data could be collected for patients treated after 2021 to explore how the recent practice guidelines have influenced physician uptake of SRS/SFRT.

## 5. Conclusions

Real-world population-based data demonstrate that post-operative brain SRS and SFRT are generally well-tolerated and effective in reducing local recurrence risk in our population-based study, consistent with clinical trial data. However, there is no evidence that SRS or SFRT confer an advantage in overall or neurological survival. A patient-centered approach weighing the risks and benefits of SRS and SFRT should be adopted. SRS or SFRT should be considered for patients who could benefit from prolonged local control. En bloc gross total resection should be considered when feasible for patients undergoing surgical resection of brain metastasis to minimize local recurrence.

## Figures and Tables

**Figure 1 curroncol-32-00345-f001:**
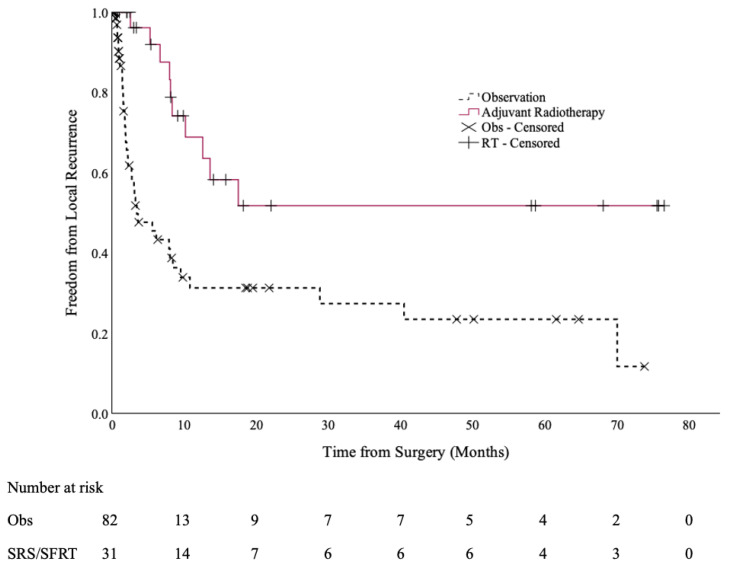
Local control of the post-operative SRS/SFRT and observation cohorts.

**Figure 2 curroncol-32-00345-f002:**
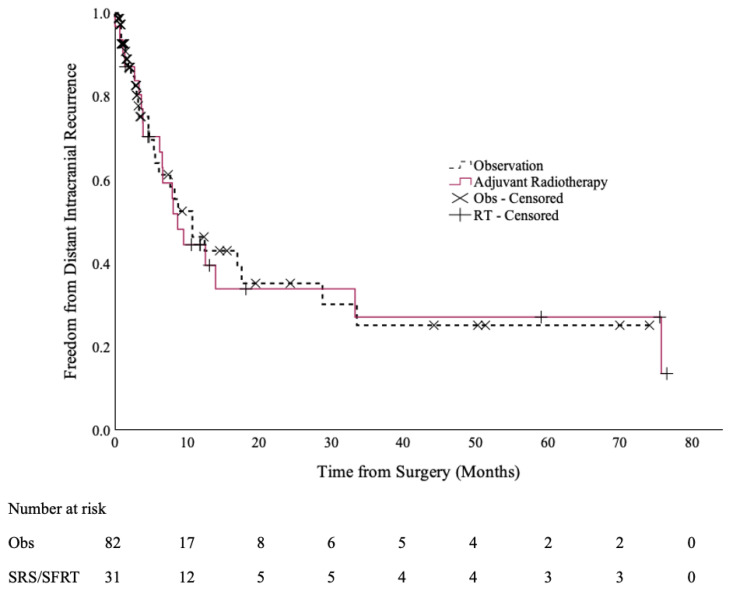
Distant intracranial control of the post-operative SRS/SFRT and observation cohorts.

**Table 1 curroncol-32-00345-t001:** Patient Baseline Characteristics.

Characteristic		SRS/SFRT; *n* (%)	Observation; *n* (%)	Significance
Number of patients		31 (27%)	82 (73%)	-
Age at brain metastasis diagnosis	<55	7 (23%)	24 (29%)	*p* = 0.4
	55–70	17 (55%)	43 (52%)	
	>70	7 (23%)	15 (18%)	
Biological sex	Female	16 (52%)	44 (54%)	*p* = 1
	Male	15 (48%)	38 (46%)	
Primary histology	Non-small cell lung cancer	18 (58%)	50 (61%)	*p* = 0.8
	Other	13 (42%)	32 (39%)	
ECOG	0–1	25 (81%)	57 (70%)	*p* = 0.3
	≥2	6 (19%)	25 (30%)	
KPS	≥70	26 (84%)	60 (73%)	*p* = 0.3
	<70	5 (16%)	22 (27%)	
Charlson Comorbidity Index (median)		8 (IQR: 7–10)	8.5 (IQR: 8–10)	*p* = 0.7
Greatest tumour dimension in cm (median)		3.4 (IQR: 2.3–4.7)	2.8 (IQR: 2.0–3.7)	*p* = 0.05
Number of brain metastases (median)		1 (IQR: 1–2)	1 (IQR: 1–2)	*p* = 0.6
Extracranial disease (ECD)	No evidence	9 (29%)	15 (18%)	*p* = 0.3
	Progressive ECD	13 (42%)	42 (51%)	
	Stable ECD	9 (29%)	25 (30%)	
En bloc tumour resection		10 (32%)	22 (27%)	*p* = 0.7
Residual disease after surgery		10 (32%)	17 (21%)	*p* = 0.3
Received prior systemic therapy		4 (13%)	23 (28%)	*p* = 0.2

**Table 2 curroncol-32-00345-t002:** Rationales for Patients not Receiving Adjuvant SRS/SFRT in the Observation Cohort.

Rationale	Number of Patients (%)
Physician preference	36 (44%)
Patient was unfit for SRS/SFRT	17 (21%)
Patient preference	11 (13%)
Radiation oncology not consulted	13 (16%)
Patient died before SRS/SFRT could be delivered	3 (4%)
Delayed SRS/SFRT, local progression in meantime	2 (2%)

**Table 3 curroncol-32-00345-t003:** Univariable and multivariable analysis of predictive factors for local control.

Variable	Univariable (Cox)		Multivariable (Cox)	
	Hazard Ratio (95% CI)	*p* Value	Hazard Ratio (95% CI)	*p* Value
Age at brain metastasis diagnosis	1.01 (0.98–1.03)	0.69		
Biological sex	0.77 (0.43–1.36)	0.36		
ECOG	1.06 (0.73–1.53)	0.78		
KPS	1.01 (0.98–1.03)	0.70		
Charlson Comorbidity Index	0.95 (0.81–1.12)	0.55		
Residual disease after surgery	1.66 (0.89–3.12)	0.097	1.95 (1.01–3.77)	0.046
Number of brain metastases	1.06 (0.89–1.26)	0.53		
Greatest tumour dimension (cm)	0.87 (0.70–1.09)	0.24		
Presence of extracranial disease (ECD)	1.52 (0.71–3.25)	0.27		
ECD progression	0.97 (0.52–1.82)	0.90		
Tumour not removed en bloc	2.09 (1.04–4.19)	0.028	2.17 (1.06–4.43)	0.034
Received post-operative SRS/SFRT	0.32 (0.16–0.64)	0.001	0.25 (0.12–0.51)	<0.001
Prior systemic treatment	1.02 (0.52–2.00)	0.90		

**Table 4 curroncol-32-00345-t004:** 12-month survival endpoints for the intent-to treat-and original analyses.

	Pseudo-ITT SRS/SFRT (95% CI)	Pseudo-ITT Observation (95% CI)	*p* Value
12-month LC	68% (49–86%)	33% (19–48%)	0.002
12-month DICC	45% (28–63%)	46% (30–63%)	0.9
12-month OS	56% (39–72%)	43% (30–55%)	0.6
12-month NS	71% (56–87%)	63% (50–76%)	0.5
	**Original SRS/SFRT (95% CI)**	**Original Observation (95% CI)**	***p* Value**
12-month LC	69% (50–88%)	31% (18–45%)	<0.001
12-month DICC	44% (26–63%)	46% (30–62%)	0.9
12-month OS	61% (44–78%)	32% (22–43%)	0.03
12-month NS	73% (56–89%)	56% (43–68%)	0.2

## Data Availability

Data are available on request due to restrictions (privacy reasons).

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
