# Peer review of "Population-Based Real-World Outcomes of Post-Operative Adjuvant Brain Cavity Radiotherapy Versus Observation"

_curroncol, 2025, doi:10.3390/curroncol32060345_

Round 1

Reviewer 1 Report

Comments and Suggestions for Authors
  1. Does the study address a clinically relevant and timely question in the management of respected brain metastases?
  2. How does this study add to or confirm existing evidence on post-operative SRS/SFRT?
  3. Is the retrospective cohort design appropriate for the research question?
  4. Were inclusion and exclusion criteria clearly defined and justified?
  5. Was the rationale behind the choice of the 2018–2020 time frame adequately explained?
  6. Was the method of patient identification via ICD-10 codes from a provincial database reliable?
  7. Were patients evenly distributed between the treatment and observation arms with minimal baseline differences?
  8. Were the definitions and delivery parameters for SRS and SFRT appropriate and consistent with guidelines?
  9. Were patients followed up long enough to observe meaningful differences in outcomes?
  10. Are the primary and secondary endpoints (LC, DICC, OS, LMD, RN) clearly defined and clinically meaningful?
  11. Is the distinction between local recurrence and radionecrosis adequately addressed in the methodology?
  12. Were Kaplan-Meier and Cox models appropriately used?
  13. Were sensitivity and ITT analyses adequately justified and interpreted?
  14. Is the reported improvement in 12-month LC (69% vs. 31%, P < 0.001) compelling enough to recommend routine post-op SRS/SFRT?
  15. The original analysis showed OS benefit, but this was lost in the ITT analysis. Does this adequately address potential selection bias?
  16. Should the authors have stratified results further by factors like tumor histology or ECOG?
  17. Is the reported rate of radionecrosis (13%, all grade 1-2) consistent with the expected literature?
  18. Was the documentation of toxicity (using CTCAE v5.0) adequate for drawing safety conclusions?
  19. Is the higher rate of LMD in the SRS/SFRT cohort (10% vs. 2%) concerning, even if statistically non-significant?
  20. Does the variability in the adoption of SRS/SFRT between 2018–2020 weaken the internal validity?
  21. Should the authors have explored reasons for lack of radiation oncology referral in 16% of cases?
  22. Are the limitations adequately discussed, particularly the small sample size and lack of quality-of-life data?
Comments on the Quality of English Language

Need to improve 

Author Response

Dear Reviewer 1,

Thank you very much for taking the time to review our manuscript. Your helpful suggestions have no doubt enhanced the quality of our manuscript. Please find the detailed responses below and the corresponding revisions/corrections highlighted/in track changes in the re-submitted files. We have also reviewed our manuscript for flow and grammar. Since we referred to some published studies in our response, we have included their citations at the end for your convenience.

Comment 1: Does the study address a clinically relevant and timely question in the management of respected brain metastases?

Response 1: Thank you for this thought-provoking question. Yes, we believe the study addresses a clinically relevant and timely question in the management of resected brain metastases. Since the publication of landmark randomized controlled trials (RCTs) by Mahajan et al. [1] and Brown et al. [2] in 2017, there has been broader adoption of immunotherapies and targeted therapies with significant intracranial efficacy. This raises the important question of whether the local control benefits associated with postoperative SRS/SFRT remain applicable in the modern therapeutic landscape. Furthermore, to our knowledge, there have been no large population-based studies evaluating the generalizability of these trial findings to a more heterogeneous, real-world patient population. Therefore, our study seeks to address this gap in the existing literature.

Comment 2: How does this study add to or confirm existing evidence on post-operative SRS/SFRT?

Response 2: Thank you for this insightful comment. First, we recognize that patients of advanced age, with multiple comorbidities, or on polypharmacy regimens, are often underrepresented in randomized controlled trials. As such, we believe that a population-based study can complement RCT findings by improving the external validity and generalizability of the existing evidence. Second, given the increasing use of systemic therapies with increased intracranial efficacy, it is important to re-evaluate whether the local control benefits of postoperative SRS/SFRT remain consistent in the contemporary setting. We have revised the Introduction accordingly to reflect these points (page 2, lines 60–67):

Therefore, integrating population-based studies with RCTs can enhance the inclusivity of the existing evidence. Notably, many patients enrolled in landmark RCTs demonstrating the local control benefit of postoperative SRS/SFRT were treated prior to the widespread adoption of immunotherapies and targeted agents with proven intracranial efficacy—such as osimertinib, alectinib, ipilimumab, and nivolumab [18-20]. Therefore, it is essential to evaluate whether the local control advantages of SRS/SFRT persist in the contemporary treatment landscape shaped by these novel systemic therapies.”

Comment 3: Is the retrospective cohort design appropriate for the research question?

Response 3: Thank you for bringing this up. We agree that prospective RCTs represent the gold standard for evaluating treatment efficacy. However, in the context of a population-based question, it is simply impossible to enroll all post-operative patients in the province to an RCT. We have also considered the possibility of a prospective cohort study, but unfortunately the most recent comprehensive list of patients who underwent surgical resection of brain metastases from the provincial surgical database is from 2020. As such, it is no longer possible to initiate prospective follow-up for that cohort. All this considered, we believe that a retrospective approach is the most pragmatic option which allows us to capture real-world data across a broader and more representative population.

Comment 4: Were inclusion and exclusion criteria clearly defined and justified?

Response 4: We are grateful for this constructive observation. We revised a sentence in the Methods section to be more clear about the inclusion and exclusion criteria (page 3, lines 94-96): “Patients were included in our study if they received surgical resection of brain metastases. Patients were excluded if they received WBRT, a total radiotherapy dose of less than 15 Gy, or if radiotherapy was delivered in more than 5 fractions.”

Because the objective of our study is to examine clinical outcomes between SRS/SFRT and observation, we did not feel compelled to include WBRT cases or patients who received a non-SRS/SFRT dose. This is explained in our Discussion (pages 11, lines 314-318).

Comment 5: Was the rationale behind the choice of the 2018–2020 time frame adequately explained?

Response 5: Thank you for this thoughtful question. We have rephrased our response in the Methods section (pages 2-3, lines 82-91): “The year 2018 was chosen as the initial cut-off point, as it follows the publication of RCTs demonstrating a local control benefit of postoperative SRS/SFRT compared to observation [8]. Patients treated after 2018 may also have been eligible for newer systemic therapies with enhanced intracranial efficacy [18-20]. The year 2020 was selected as the end point, as it precedes the release of consensus guidelines formally recommending the use of postoperative SRS/SFRT [13,14]. This interval represents a transitional period during which we would expect high variability in physician adoption of postoperative SRS/SFRT, which provides a unique opportunity to examine the real-world outcomes of post-operative SRS/SFRT compared to observation.”

In addition, we also edited a portion of the Discussion to further emphasize our point (page 10, lines 289-293): “Study inclusion years between 2018 – 2020 was specifically chosen to compare the outcome between post-operative SRS/SRT and observation while minimizing baseline characteristic discrepancies between the two subset and to allow for sufficiently long follow up. Between 2018-2020, uptake of SRS/SFRT among treating physicians was variable as the guidelines recommending routine post-op SRS was published after 2020 [13-15].

Comment 6: Was the method of patient identification via ICD-10 codes from a provincial database reliable?

Response 6: Thank you for bringing this up. We obtained patient records from the provincial Discharge Abstract Database (DAD), which captures clinical information from all acute inpatient care facilities in British Columbia. Since 2004–2005, all DAD entries have been coded using the ICD-10 classification system. To identify our study cohort, we specifically requested records for patients diagnosed with "secondary malignant neoplasm of the brain and cerebral meninges" (ICD-10 code: C79.3). This would in theory encompass all patients diagnosed with brain metastases across the province. The DAD records came with a list of neurosurgical interventions received by each patient. We then filtered this dataset to include only patients who underwent surgical resection for brain metastases. To ensure accuracy, we conducted a manual chart review of each case in the final dataset to confirm that surgical resection was indeed performed for brain metastasis. We revised a sentence in the Methods section to better clarify our approach: “Patients were identified through the provincial Ministry of Health Discharge Abstract Database by selecting for those with an ICD-10 diagnosis code of C79.3 and a documented history of neurosurgical intervention." (page 2, lines 80-82).

Comment 7: Were patients evenly distributed between the treatment and observation arms with minimal baseline differences?

Response 7: As shown in Table 1, the baseline characteristics between the two patient cohorts were largely comparable. We acknowledge that the SRS/SFRT cohort included fewer patients, which may limit statistical power. Nevertheless, this imbalance in cohort sizes is a common feature of retrospective studies, and it is reflective of the real-world distribution of treatment decisions during the study period.

Comment 8: Were the definitions and delivery parameters for SRS and SFRT appropriate and consistent with guidelines?

Response 8: This is an excellent question. Across BC Cancer (a multi-centre institution serving the entirety of British Columbia), our dosage/fractionation for SRS and SFRT during 2018-2020 were 18-24 Gy in 1 fraction and 30-35 Gy in 5 fractions, respectively. There is no expansion for the CTV. The PTV is a uniform 1 mm expansion of the GTV. Treatment plans were optimized to aim for Dmax < 150%, with Dmax situated within the GTV, and PTV100% > 99%. Whenever possible, pre-radiotherapy MRI with ≤1 mm slice thickness was obtained. Radiotherapy was delivered using 6 MV photons with the VMAT technique by linear accelerators. Patients were immobilized with a thermoplastic shell, and image verification with cone beam CT was performed. This is consistent with published guidelines on brain SRS/SFRT [3,4]. We have included all this information in the Methods section (page 3, lines 128-133).

Comment 9: Were patients followed up long enough to observe meaningful differences in outcomes?

Response 9: We appreciate this inquisitive comment. Because our patient inclusion period was from 2018-2020, and our last chart review update was in October 2024, patients would have received at least 46 months (nearly 4 years) of follow up. While a small portion of patients are long-term survivors, most do not survive beyond 4 years after diagnosis of brain metastasis. In our study, 94 patients (83%) died by the time of our last chart update. Therefore, we believe the patients in our study have received sufficient length of follow up.

Comment 10: Are the primary and secondary endpoints (LC, DICC, OS, LMD, RN) clearly defined and clinically meaningful?

Response 10: Thank you for highlighting this aspect of our study. We defined local recurrence as “radiographic evidence of any new or progressive contrast-enhancing nodularity contiguous to or within the tumour cavity.” This is consistent with a previous RCT by Mahajan et al [1]. We have slightly modified the definition of distant intracranial progression as “time from surgery to the appearance of a new or progressive lesion on imaging that was distinct from the tumour cavity” (page 3, lines 108-109) to clarify that this is a radiological diagnosis [1]. The overall survival endpoint is quite objective (i.e. patient death), so we did not further elaborate on this. For neurologic survival, patients who died with “progressive or severe neurologic dysfunction were considered to have died of neurologic causes, regardless of whether their systemic cancer or any intercurrent disease(s) were also contributory; deaths were non-neurologic if the patient was stable or improving from a neurologic perspective at the time of death.” This definition is in accordance with another previous RCT by Patchell et al [5]. In our study, leptomeningeal disease (LMD) was diagnosed based on radiological findings, except in cases where malignant cells were identified through cerebrospinal fluid analysis [2]. Finally, radionecrosis is also determined radiologically, and if applicable, through pathology reports [6]. We have rephrased the corresponding sentences in the Methods section (page 3, lines 108-111): “LMD was assessed based on radiology reports, and when available, cerebrospinal fluid analysis. Radionecrosis was evaluated using radiology reports and supplemented by pathology findings, if applicable.”

Comment 11: Is the distinction between local recurrence and radionecrosis adequately addressed in the methodology?

Response 11: Thank you for drawing our attention to this nuanced topic. Differentiating between local recurrence and radionecrosis on imaging can be challenging. Because lesions that appear suspicious are not usually biopsied, pathology reports are rarely available to provide a definitive answer. At our institution, ambiguous cases are brought to discussion at a multidisciplinary tumour board, where multiple radiologists and oncologists can provide their expert opinion. This is now reflected in the Methods section (page 3, lines 111-113): “In situations where distinction between local recurrence and radionecrosis was unclear, the cases were reviewed at multidisciplinary tumour board meetings, and the final diagnosis would be determined by group consensus.

Comment 12: Were Kaplan-Meier and Cox models appropriately used?

Response 12: We appreciate this astute question. Our study was performed with the guidance of a trained statistician (one of our co-authors), who provided input on the appropriate statistical methods to use. Following our discussion, we determined that the Kaplan-Meier model was suitable for survival analysis, and the Cox proportional hazards model was appropriate for both uni- and multivariable analysis.

Comment 13: Were sensitivity and ITT analyses adequately justified and interpreted?

Response 13: Thank you for this constructive critique. Given the retrospective nature of our study, there is a potential for selection bias, as many patients in the observation cohort were not suitable candidates for SRS/SFRT. In addition, 5 patients were initially planned to receive SRS/SFRT but either died or experienced local recurrence before radiotherapy could be delivered. After discussion with our statistician co-author, we decided to conduct a sensitivity analysis to assess whether the observed difference in overall survival was truly associated with SRS/SFRT and to evaluate whether the local control benefit remained consistent even if those 5 patients were reassigned to the SRS/SFRT cohort. We acknowledge that because this is not a randomized trial, referring to the analysis as “ITT” would be inappropriate. Therefore, we have revised our manuscript to refer to the sensitivity analysis as “pseudo-ITT.” We explained our interpretation of the sensitivity analysis in the Discussion (page 10, lines 275-280): “Interestingly, in our original analysis there was an overall survival (OS) benefit with post-operative SRS/SFRT. However, this is most likely due to selection bias, as the sensitivity analysis revealed no OS benefit... This is supported by the finding that neither the original nor the sensitivity analyses demonstrated a difference in neurological survival (NS).”

Comment 14: Is the reported improvement in 12-month LC (69% vs. 31%, P < 0.001) compelling enough to recommend routine post-op SRS/SFRT?

Response 14: Thank for you this excellent question. The magnitude of improvement in local control is 38% (35% if we look at the pseudo-ITT analysis). This is slightly higher than the 29% difference found by Mahajan et al [1]. We acknowledge that there is no overall survival benefit on sensitivity analysis, and similarly no neurologic survival benefit is demonstrated. We have emphasized in the conclusion (page 11, lines 329-331) that “[a] patient-centred approach weighing the risks and benefits of SRS and SFRT should be adopted.”

Comment 15: The original analysis showed OS benefit, but this was lost in the ITT analysis. Does this adequately address potential selection bias?

Response 15: Thank you for this astute observation. As you correctly pointed out, the OS benefit in the primary analysis was lost in the pseudo-ITT analysis. This is because 17 patients who were not clinically fit for SRS/SFRT were excluded, and another 5 patients who were initially intended to receive SRS/SFRT (but did not ultimately undergo treatment) were reassigned from the observation cohort to the SRS/SFRT cohort. As with any retrospective review, there is an inherent risk of selection bias, which we sought to mitigate through our pseudo-ITT analysis. Given that the pseudo-ITT analysis did not demonstrate an OS benefit, we believe our approach adequately addresses the risk of selection bias, and that further sensitivity analyses would be unlikely to change our conclusions regarding OS. Nevertheless, we would be pleased to consider any specific post hoc analyses that you may suggest to further reduce the risk of selection bias.

Comment 16: Should the authors have stratified results further by factors like tumor histology or ECOG?

Response 16: Thank you very much for raising this important point. We believe that stratifying for tumour histology is important, because new systemic agents for non-small cell lung cancer (NSCLC) such osimertinib and alectinib have been shown to have robust intracranial efficacy. We found that among patients with NSCLC, 12-month local control following SRS/SFRT is 65% (95% CI: 37-93%), which is higher than observation (24%, 95% CI: 7-41%; P = 0.001). Distant intracranial control is not significantly different between the two cohorts (P = 0.447), and neither is overall survival (P = 0.06). We have mentioned the NSCLC LC results on page 6, lines 183-185: “Specifically, for patients with non-small cell lung cancer (NSCLC), their 12-month LC was 65% (95% CI: 37-93%) if they received SRS/SFRT, compared to 24% (95% CI: 7-41%) if they went on observation (P = 0.001).”

Unfortunately, we did not have enough sample size for the other tumour histologies to perform further analyses. We also found that when patients with ECOG of 2 or higher were excluded from the OS analysis, the OS difference between the two cohorts disappeared (P = 0.29). We have included this finding in the Results (page 7, lines 197-199): “When patients with an ECOG of 2 or higher were excluded from the OS analysis, the 12-month OS was 60% (95% CI: 41-79%) in the SRS/SFRT cohort and 40% (95% CI: 27-52%) in the observation cohort (P = 0.29).”

Comment 17: Is the reported rate of radionecrosis (13%, all grade 1-2) consistent with the expected literature?

Response 17: We appreciate this inquisitive comment. A recent literature review examined 13 published series on SFRT and found the mean rate of radionecrosis to be 10.3% (0-20.6%) [7]. Therefore, we believe that our reported rate of 13% is consistent with the published literature.

Comment 18: Was the documentation of toxicity (using CTCAE v5.0) adequate for drawing safety conclusions?

Response 18: Thank you for bringing up this topic. As this is a retrospective study, we were limited to reviewing the patients’ charts for evidence of documented toxicity, as opposed to having standardized documentation of toxicity at each follow up visit. We have tempered our Discussion (page 10, lines 286-288) to state that “[g]iven the retrospective design of our study, prospective toxicity data were not available. However, retrospective review revealed no documented grade 3 or higher toxicities.

Comment 19: Is the higher rate of LMD in the SRS/SFRT cohort (10% vs. 2%) concerning, even if statistically non-significant?

Response 19: Thank you for this valuable and thought-provoking question. Although the LMD rate appears higher in the SRS/SFRT cohort, this only represents 3 patients. It is likely that the observed difference is due to random variation rather than a true correlation, hence why it is not statistically significant, as you correctly pointed out. Published literature reports that post-SRS LMD rates can be approximately 15% [8], so our observed rate of 10% is not unexpected. Therefore, we do not consider this finding to be particularly concerning.

Comment 20: Does the variability in the adoption of SRS/SFRT between 2018–2020 weaken the internal validity?

Response 20: Thank you for this excellent question. While we acknowledge that internal validity may be somewhat compromised by this, it is well-established that RCTs are the gold standard for internal validity and establishing causality, and previous randomized evidence has already demonstrated a local control benefit with SRS [1]. Hence, the main goal of our study is not to replicate the internal validity of previous RCTs, but rather to assess their external validity in a real-world, population-based setting.

Comment 21: Should the authors have explored reasons for lack of radiation oncology referral in 16% of cases?

Response 21: Thank you for bringing this to our attention. Unfortunately, in most cases, the rationale behind not referring a patient to radiation oncology is not explicitly stated. Due to the retrospective nature of our review, we are limited to the information that is available in the patients’ charts. Not all physicians include their reasoning in their dictations, and often the dictations may be located on private electronic medical record systems, which we cannot access.

Comment 22: Are the limitations adequately discussed, particularly the small sample size and lack of quality-of-life data?

Response 22: Thank you and we welcome this insightful critique. We have rephrased this part of the Discussion as below, with the edited parts in red (pages 10-11, lines 305-314): “One limitation to our study is the small sample of SRS/SFRT cases which limits analyses stratified for performance status and tumour histology, and detailed dosimetric analysis to evaluate additional predictive factors for local control. Similarly, the numbers of cases with leptomeningeal recurrence and radionecrosis were low and not sufficient for any exploratory analyses. In addition, as this was a retrospective study, another limitation is the absence of quality-of-life data. This precludes assessment of patient-reported effects of SRS/SFRT. Understandably, without data demonstrating improved quality-of-life or a significant neurologic survival benefit, some patients may not wish to receive SRS/SFRT and some physicians may not feel inclined to offer it.

Thank you again for your review and considering our submission. Please do not hesitate to let us know if you have further comments or suggestions for our manuscript.

References:

  1. Mahajan A, Ahmed S, McAleer MF, Weinberg JS, Li J, Brown P, Settle S, Prabhu SS, Lang FF, Levine N, et al. Post-operative stereotactic radiosurgery versus observation for completely resected brain metastases: a single-centre, randomised, con-trolled, phase 3 trial. Lancet Oncol. 2017 Aug 1;18(8):1040–8.
  2. Brown PD, Ballman KV, Cerhan JH, Anderson SK, Carrero XW, Whitton AC, Greenspoon J, Parney IF, Laack NN, Ashman JB, et al. Postoperative stereotactic radiosurgery compared with whole brain radiotherapy for resected metastatic brain disease (NCCTG N107C/CEC·3): a multicentre, randomised, controlled, phase 3 trial. Lancet Oncol. 2017 Aug 1;18(8):1049–60.
  3. Gondi V, Bauman G, Bradfield L, Burri SH, Cabrera AR, Cunningham DA, Eaton BR, Hattangadi-Gluth JA, Kim MM, Ko-techa R, et al. Radiation Therapy for Brain Metastases: An ASTRO Clinical Practice Guideline. Pract Radiat Oncol. 2022 Jul 1;12(4):265–82.
  4. Hartgerink D, Swinnen A, Roberge D, Nichol A, Zygmanski P, Yin FF, Deblois F, Hurkmans C, Ong CL, Bruynzeel A, Aizer A. LINAC based stereotactic radiosurgery for multiple brain metastases: guidance for clinical implementation. Acta Oncologica. 2019 Sep 2;58(9):1275-82.
  5. Patchell RA, Tibbs PA, Regine WF, Dempsey RJ, Mohiuddin M, Kryscio RJ, Markesbery WR, Foon KA, Young B. Postopera-tive Radiotherapy in the Treatment of Single Metastases to the Brain - A Randomized Trial. JAMA. 1998 Nov 4;280(17):1485–9.
  6. Doré M, Martin S, Delpon G, Clément K, Campion L, Thillays F. Stereotactic radiotherapy following surgery for brain me-tastasis: Predictive factors for local control and radionecrosis. Cancer/Radiothérapie. 2017 Feb 1;21(1):4–9.
  7. Rogers S, Stauffer A, Lomax N, Alonso S, Eberle B, Gomez Ordoñez S, Lazeroms T, Kessler E, Brendel M, Schwyzer L, et al. Five fraction stereotactic radiotherapy after brain metastasectomy: a single-institution experience and literature review. J Neurooncol. 2021 Oct 1;155(1):35–43.
  8. Shi S, Sandhu N, Jin MC, Wang E, Jaoude JA, Schofield K, Zhang C, Liu E, Gibbs IC, Hancock SL, et al. Stereotactic Radio-surgery for Resected Brain Metastases: Single-Institutional Experience of Over 500 Cavities. Int J Radiat Oncol. 2020 Mar 15;106(4):764–71.

Reviewer 2 Report

Comments and Suggestions for Authors

This is an interesting study that reaffirms improved local control with postoperative SRS/SFRT in a real-world setting. However, the novelty is limited, and the manuscript would benefit from the following clarifications:

i) Limited Novelty: The benefit of postoperative SRS/SFRT is already well established. Please clarify the unique contribution of your study beyond existing evidence.

ii) Use of “ITT”: Since this is a retrospective study, the term “intention-to-treat” may be misleading. We recommend using “pseudo-ITT” or “intended treatment analysis” for accuracy.

iii) QOL Consideration: As SRS/SFRT is often chosen to preserve neurologic function and quality of life, the absence of QOL data should be acknowledged and discussed as a limitation.

iv) Equipment Details: Please specify the type of radiotherapy device used (e.g., LINAC, Gamma Knife, CyberKnife), as this affects reproducibility and interpretation.

These revisions would enhance the clarity and scientific value of the manuscript.

Author Response

Comment 1: This is an interesting study that reaffirms improved local control with postoperative SRS/SFRT in a real-world setting. However, the novelty is limited, and the manuscript would benefit from the following clarifications:

  1. i) Limited Novelty: The benefit of postoperative SRS/SFRT is already well established. Please clarify the unique contribution of your study beyond existing evidence.

Response 1: Thank you for bringing up this pertinent point. We acknowledge that the local control benefits of SRS/SFRT are well established through randomized controlled trials (RCTs). However, we believe that our research complements the existing evidence by confirming the external validity of these RCTs on a population basis. In drug RCTs, over a quarter of patients over the age of 60 and over half of those over the age of 80 were excluded [1]. Therefore, there is value in demonstrating that results from RCTs remain applicable to the more heterogenous and comorbid general population. We have emphasized the population-based nature of our research in the Abstract (page 1, lines 31-33) “Post-operative SRT outcomes based on real-world population data are consistent with the data from clinical trials and support the established guidelines” and Conclusion (page 11, lines 326-328) “Real-world population-based data demonstrate that post-operative brain SRS and SFRT are generally well-tolerated and effective in reducing local recurrence risk in our population-based study, consistent with clinical trial data.”

Another novel aspect of our study is that it was conducted in the modern era where more CNS-penetrating targeted therapies and immunotherapies were widely available. We have emphasized these above points in the Introduction (page 2, lines 60-67): “Therefore, integrating population-based studies with RCTs can enhance the inclusivity of the existing evidence. Notably, many patients enrolled in landmark RCTs demonstrating the local control benefit of postoperative SRS/SFRT were treated prior to the widespread adoption of immunotherapies and targeted agents with proven intracranial efficacy—such as osimertinib, alectinib, ipilimumab, and nivolumab [18-20]. Therefore, it is essential to evaluate whether the local control advantages of SRS/SFRT persist in the contemporary treatment landscape shaped by these novel systemic therapies.”

Our findings that SRS/SFRT retained their effectiveness in improving local control (LC) gives us further confidence that the current practice guidelines remain valid in the immunotherapy era. We have emphasized this in the Discussion (page 10, line 270-274): “Notably, the local control benefit of post-operative SRS/SFRT remains consistent even though approximately one-third of patients received targeted or immunotherapy. This supports the continued relevance of current practice guidelines in the context of modern CNS-penetrating systemic treatments.”

Comment 2: ii) Use of “ITT”: Since this is a retrospective study, the term “intention-to-treat” may be misleading. We recommend using “pseudo-ITT” or “intended treatment analysis” for accuracy.

Response 2: Thank you for this helpful comment. We agree that it would be more methodologically accurate to call this an “intended treatment analysis”. We have changed all instances of “ITT” into “pseudo-ITT” and highlighted them in our revised manuscript in accordance with your recommendations. 

Comment 3: iii) QOL Consideration: As SRS/SFRT is often chosen to preserve neurologic function and quality of life, the absence of QOL data should be acknowledged and discussed as a limitation.

Response 3: Thank you for pointing this out. We have modified a section in the Discussion (pages 10-11, lines 310-314) to clarify that the lack of QOL data is a limitation to the study. This has been highlighted in the manuscript for your convenience.

In addition, as this was a retrospective study, another limitation is the absence of quality-of-life data. This precludes assessment of patient-reported effects of SRS/SFRT. Understandably, without data demonstrating improved quality-of-life or a significant neurologic survival benefit, some patients may not wish to receive SRS/SFRT and some physicians may not feel inclined to offer it.

Comments 4: iv) Equipment Details: Please specify the type of radiotherapy device used (e.g., LINAC, Gamma Knife, CyberKnife), as this affects reproducibility and interpretation.

Response 4: Thank you for this attentive comment. All brain SRS/SFRT patients were treated with linear accelerators. We have included more details on our treatment protocol on page 3, lines 128-133: “Treatment plans were optimized to aim for Dmax < 150%, with Dmax situated within the GTV, and PTV100% > 99%. Whenever possible, pre-radiotherapy MRI with ≤1 mm slice thickness was obtained. Radiotherapy was delivered using 6 MV photons with the VMAT technique by linear accelerators. Patients were immobilized with a thermoplastic shell, and image verification with cone beam CT was performed. This is consistent with published guidelines on brain SRS/SFRT.

References:

  1. Tan YY, Papez V, Chang WH, Mueller SH, Denaxas S, Lai AG. Comparing clinical trial population representativeness to real-world populations: an external validity analysis encompassing 43 895 trials and 5 685 738 individuals across 989 unique drugs and 286 conditions in England. The Lancet Healthy Longevity. 2022 Oct 1;3(10): e674-89.

Reviewer 3 Report

Comments and Suggestions for Authors

This is a clearly presented and timely retrospective population-based study evaluating postoperative SRS/SFRT versus observation in patients undergoing surgery for brain metastases. The authors provide compelling real-world evidence that postoperative cavity-directed radiotherapy significantly improves local control compared to observation, replicating RCT findings in a more inclusive general population. The analysis is well-structured and appropriately performed, with thoughtful use of multivariable modeling and a justified intent-to-treat sensitivity analysis.

A few points merit emphasis. First, the absence of a survival benefit, despite improved local control, is consistent with existing randomized data, but still highlights the importance of a patient-centered approach, particularly in patients with limited expected survival. The authors appropriately acknowledge that selection bias likely accounts for the observed OS advantage in the initial per-protocol analysis. The sensitivity analysis successfully mitigates this.

Second, the study’s strength lies in its broad inclusion of real-world patients and detailed reporting of reasons for non-referral or treatment omission. These add practical insights for implementation of postoperative radiotherapy strategies. The relatively small number of SRS/SFRT cases limits deeper dosimetric exploration, but this is appropriately framed.

Finally, the toxicity profile is consistent with expectations, and the discussion appropriately tempers interpretation of the LMD findings given the small event numbers.

Overall, this is a strong contribution that confirms the external validity of guideline-based postoperative radiotherapy for brain metastases and highlights gaps in uptake during the transition period. No major revisions are required.

Author Response

Comments 1: 

This is a clearly presented and timely retrospective population-based study evaluating postoperative SRS/SFRT versus observation in patients undergoing surgery for brain metastases. The authors provide compelling real-world evidence that postoperative cavity-directed radiotherapy significantly improves local control compared to observation, replicating RCT findings in a more inclusive general population. The analysis is well-structured and appropriately performed, with thoughtful use of multivariable modeling and a justified intent-to-treat sensitivity analysis.

A few points merit emphasis. First, the absence of a survival benefit, despite improved local control, is consistent with existing randomized data, but still highlights the importance of a patient-centered approach, particularly in patients with limited expected survival. The authors appropriately acknowledge that selection bias likely accounts for the observed OS advantage in the initial per-protocol analysis. The sensitivity analysis successfully mitigates this.

Second, the study’s strength lies in its broad inclusion of real-world patients and detailed reporting of reasons for non-referral or treatment omission. These add practical insights for implementation of postoperative radiotherapy strategies. The relatively small number of SRS/SFRT cases limits deeper dosimetric exploration, but this is appropriately framed.

Finally, the toxicity profile is consistent with expectations, and the discussion appropriately tempers interpretation of the LMD findings given the small event numbers.

Overall, this is a strong contribution that confirms the external validity of guideline-based postoperative radiotherapy for brain metastases and highlights gaps in uptake during the transition period. No major revisions are required.

Response 1: 

Thank you for your thorough review and affirmative evaluation of our research. We are very pleased to hear that you believe it is a strong contribution to the topic of postoperative radiotherapy for brain metastases. As no revisions were suggested, we did not make any changes to the manuscript in response to your thoughtful review. If you have any further comments in the future, we would be happy to address them. Once again, we greatly appreciate your encouraging comments.

Round 2

Reviewer 1 Report

Comments and Suggestions for Authors

Author carry over all the comments

Comments on the Quality of English Language

Good

Reviewer 2 Report

Comments and Suggestions for Authors

I would like to note that the authors have responded thoroughly and constructively to my comments. The revised manuscript reflects a careful consideration of the suggestions, particularly in addressing potential selection bias, clarifying the limitations of the study, and improving the clarity of the statistical methods and conclusions. The updates enhance both the scientific rigor and clinical relevance of the manuscript.